# Thioester-Containing Protein TEP27 in *Culex quinquefasciatus* Promotes JEV Infection by Modulating Host Immune Function

**DOI:** 10.3390/ijms262311727

**Published:** 2025-12-03

**Authors:** Yutian Huang, Yuwei Liu, Rongrong Li, Xi Zhu, Ruidong Li, Sihao Peng, Xin An, Yuxin Yang, Yuanyuan Liu, Yiping Wen, Qin Zhao, Shan Zhao, Fei Zhao, Rui Wu, Xiaobo Huang, Qigui Yan, Yifei Lang, Yiping Wang, Yajie Hu, Yi Zheng, Sanjie Cao, Senyan Du

**Affiliations:** 1Research Center for Swine Diseases, College of Veterinary Medicine, Sichuan Agricultural University, Chengdu 611130, China; huangyutian0526@163.com (Y.H.); 13872789583@163.com (Y.L.); 15682556823@163.com (R.L.); 15283276421@163.com (X.Z.); liruidong764@163.com (R.L.); 18884737596@163.com (S.P.); anxin29292x@163.com (X.A.); yyx2277@163.com (Y.Y.); 18373980318@163.com (Y.L.); wyp@sicau.edu.cn (Y.W.); zhao.qin@sicau.edu.cn (Q.Z.); zhaoshan419@163.com (S.Z.); zhaofei100@yeah.net (F.Z.); wurui1977@sicau.edu.cn (R.W.); rsghb110@126.com (X.H.); yanqigui@126.com (Q.Y.); y_langviro@163.com (Y.L.); 14918@sicau.edu.cn (Y.W.); zhengyi132@sicau.edu.cn (Y.Z.); csanjie@sicau.edu.cn (S.C.); 2Sichuan Science-Observation Experimental Station for Veterinary Drugs and Veterinary Diagnostic Technology, Ministry of Agriculture, Chengdu 611130, China; 3Engineering Research Center of Southwest Animal Disease Prevention and Control Technology, Ministry of Education of the People’s Republic of China, Chengdu 611130, China; 4Key Laboratory of Animal Disease and Human Health of Sichuan Province, Sichuan Agricultural University, Chengdu 611130, China; 5Sichuan Center for Disease Control and Prevention, Chengdu 610041, China; hengqiaojie1984@163.com

**Keywords:** *Culex quinquefasciatus*, thioester-containing protein (TEP), *Japanese encephalitis virus*, innate immunity, antimicrobial peptides

## Abstract

Thioester-containing proteins (TEPs), which are distinguished by the thioester motif (GCGEQ), are essential to arthropods’ defense against infections. Although TEPs have been extensively investigated in *Anopheles*, *Aedes*, and *Drosophila*, their functions in *Culex* mosquitoes remain inadequately explored. Interestingly, we discovered that *Culex* TEPs exhibit functional antagonism to their orthologs in other species, actively facilitating viral infection in this vector. In this study, we identified nine *TEP* genes in *Culex quinquefasciatus*, three of which were found to critically facilitate *Japanese encephalitis virus* (JEV) infection, with *CqTEP27* exhibiting the most pronounced proviral effect. Mechanistically, *Cq*TEP27 may have suppressed the production of several antimicrobial peptides (AMPs), which increased JEV replication. Our work also highlights the potential of targeting susceptibility factors such as *Cq*TEP27 to block pathogen acquisition. Notably, the rate of mosquito infection was significantly decreased by membrane blood feeding antisera against *Cq*TEP27. Therefore, vaccination against *Cq*TEP27 offers a workable method of avoiding JEV infection. According to our research, *Cq*TEP27 is a promising target for the development of vaccines that prevent JEV transmission. By preventing viral infection in mosquitoes that feed on immunized hosts, this approach can directly disrupt the natural transmission cycle, offering a novel strategy to reduce the disease burden.

## 1. Introduction

In Southeast Asia and the Western Pacific, the most common cause of viral encephalitis in humans is still the *Japanese encephalitis virus* (JEV), a mosquito-borne flavivirus of major global health concern [1]. Largely transmitted by *Culex* mosquitoes (*C. tritaeniorhynchus* and *C. quinquefasciatus*), JEV has recently expanded its geographic range beyond its traditional endemic areas, such as China, India, and Southeast Asia, into regions including Pakistan, Africa, and Australia [2]. This geographical expansion is driven primarily by climate change, urban development, and the movement of animal populations [3]. The spread of JEV is a significant public health concern, as it can lead to severe encephalitis cases, which are characterized by high mortality (20–30%) and significant disability rates (30–50%) [4]. Concurrently, JEV adversely impacts food security by inducing abortion storms and infertility in swine populations. Infected pigs exhibit high-titer viremia with age-dependent pathological outcomes: piglets often succumb to fatal nonsuppurative encephalitis, whereas breeding adults suffer abortions and infertility, severely disrupting swine production [5]. Consequently, the spread of JEV jeopardizes both public health and food security.

The transmission cycle of JEV heavily relies on pigs, which act as primary amplifying hosts [6]. Female mosquitoes acquire the virus during blood-feeding on viremic hosts, and following a 10–14-day extrinsic incubation period, the virus disseminates to the salivary glands, enabling transmission to new hosts [7]. Mosquitoes also serve as crucial environmental reservoirs by facilitating transovarial transmission, thereby ensuring viral persistence during inter-epidemic periods [8]. The efficacy of current JEV vaccines, which are predominantly derived from the Genotype III (GIII) strain is well established [9,10]. Nonetheless, their effectiveness against emerging and divergent genotypes like GI and GV remains uncertain, raising questions about the breadth of protection in the face of increasing viral genetic diversity [11]. The lack of antiviral treatments, combined with low vaccination rates in areas of expansion and ineffective barriers to mosquito-pig transmission, necessitates the development of new strategies to disrupt the enzootic cycle [12].

In arthropods, which lack immunoglobulin-mediated adaptive immunity, the innate immune system serves as the foremost defense mechanism protecting against viral infections [13]. This system is fundamentally dependent on pattern recognition receptors (PRRs), which identify pathogen-associated molecular patterns (PAMPs) through conserved molecular interactions [14]. The major PRRs identified in arthropods include peptidoglycan recognition proteins (PGRPs), Gram-negative bacteria-binding proteins (GNBPs), Lectins, and thioester-containing proteins (TEPs) [15]. Among these, TEPs constitute an evolutionarily conserved protein family that spans from protostomes to deuterostomes, exhibiting remarkable structural and functional conservation [16]. These immune effectors are characterized by a signature thioester motif (GCGEQ), where a thioester bond forms between the cysteine thiol group and the glutamine acyl group—a molecular feature shared with the vertebrate complement component C3 [17]. The structural homology of TEPs underpins their analogous functions in innate immunity, where they are involved in pathogen opsonization, developmental regulation, and host–microbe interactions [18]. Phylogenetic and functional analyses categorize the TEPs superfamily into four groups: Alpha-2 macroglobulin (A_2_M), C3-like complement, insect TEPs (iTEPs), and macroglobulin complement-related protein (MCR) [18]. This classification underscores their diverse yet specialized roles in immune surveillance and effector mechanisms. Extensive research has demonstrated that TEPs have multifaceted roles in insect immunity, mediating defense responses against a variety of pathogens, including bacterial phagocytosis, plasmodium lysis, and fungal melanization [19,20,21]. In *Anopheles* mosquitoes, the functional characterization of TEPs reveals conserved immunological mechanisms. Specifically, *Ag*TEP1, one of the most extensively studied and functionally central TEPs, secreted into the hemolymph, functionally parallels vertebrate complement C3 by opsonizing fungi, Gram-positive and Gram-negative bacteria for phagocytic clearance [22,23,24]. Beyond bacterial defense, *Ag*TEP1 exhibits potent anti-plasmodium activity through its interaction with the LRIM1/APL1 complex, facilitating direct binding to parasite surfaces and initiating lytic and melanization pathways [23]. TEPs have been extensively studied in *Anopheles* mosquitoes, and similarly, their antiviral capabilities have been roughly studied in *Aedes aegypti*. Despite lacking the conserved thioester motif, the macroglobulin complement-related factor (*Aa*MCR) shows broad inhibitory effects on various flaviviruses by prompting the generation of antimicrobial peptides (AMPs) [25]. Similarly, *Aa*TEP20 enhances resistance to dengue virus type 2 (DENV2), Zika virus, and chikungunya (CHIKV) virus by regulating the JNK pathway in the salivary glands [26]. Functional studies further demonstrate that silencing *AaTEP1* and *AaTEP3* increased West Nile virus (WNV) accumulation by 2–3 fold; in contrast, ectopic expression of *AaTEP1* suppressed viral replication, thereby confirming their roles as viral restriction factors [27]. Despite these advances in *Anopheles* and *Aedes* systems, the immunological functions of TEPs in *C. quinquefasciatus* remain inadequately explored, with their potential antiviral mechanisms against significant arboviruses yet to be characterized.

Vector-derived factors that mediate arbovirus acquisition represent promising targets for disrupting viral transmission cycles. This principle has been previously demonstrated in *Ae. aegypti*, where inhibition of mosquito C-type lectin 3 (GCTL-3), a PRR like TEPs, functions in innate immune recognition, significantly reduces DENV acquisition [28,29]. Building on this concept, we conducted a systematic functional screening of the TEP family in *C. quinquefasciatus* to identify potential regulators of JEV infection. Through in vivo RNA interference (RNAi) screening of all nine *TEP* family genes, we identified three that significantly influence JEV infectivity, with *Cq*TEP27 emerging as the most potent proviral factor. Passive immunization with *Cq*TEP27-specific antisera during blood feeding markedly reduced JEV acquisition, which confirms that its extracellular domains are accessible to antibodies. Mechanistically, our findings demonstrate that *Cq*TEP27 suppresses the expression of AMPs, thereby promoting JEV replication. Importantly, an immunization strategy targeting *Cq*TEP27 could potentially reduce the prevalence of infected mosquitoes.

These findings establish *Cq*TEP27 as a viable target for a transmission-blocking vaccine, offering a strategic approach to interrupt the JEV enzootic cycle. By preventing viral uptake at the vector level, such interventions could complement existing JEV control measures.

## 2. Results

### 2.1. The Function of the CqTEP Family in C. quinquefasciatus Infections

TEPs are phylogenetically divided into four major superfamilies: A_2_M, C3-like complement, and insect-specific iTEPs, and MCR [18]. TEPs contain the thioester (TE) motif, GCGEQ, featuring a highly unstable covalent bond between the side chains of cysteine and a neighboring glutamic acid [17]. Utilizing these protein sequence characteristics and homologous genes from other species, we identified nine TEP homologous genes in *C. quinquefasciatus*. Subsequent domain architecture analysis using SMART revealed that most TEP contain several characteristic domains (A2M_N, A2M, Thiol-ester_cl (TED), A2M_comp, and A2M_recep). To validate the phylogenetic classification and infer functions of *Culex TEPs*, we constructed a phylogenetic tree using orthologs from key dipteran species: the model organism *Drosophila melanogaster* and the mosquito vectors *Aedes aegypti* and *Anopheles gambiae*. This evolutionary framework reliably delineates orthology relationships and enables the formulation of testable hypotheses regarding the roles of specific *Culex* TEPs in JEV infection. As illustrated in Figure 1A, the majority of the TEPs in *C. quinquefasciatus* belongs to the iTEP group, with only *Cq*TEP13 classified within the MCR family. Previous studies have demonstrated that *Ag*TEP1 specifically recognizes *Plasmodium ookinetes* [30,31], mediates their lysis, and acts as an opsonin to promote phagocytosis of Gram-negative and Gram-positive bacteria [21,24,32]. Likewise, the orthologous proteins in *D. melanogaster*, such as *Dm*TEP2, *Dm*TEP3, and *Dm*TEP6, also function as opsonins that bind to both fungi and bacteria, thereby markedly enhancing the phagocytic efficiency of hemocytes [19]. Based on the conserved characteristic domains present in most *Culex* TEPs, we therefore hypothesize that the TEP gene family in *C. quinquefasciatus* may perform a comparable function in regulating microbial infections. To assess the function of TEPs in JEV infection, mosquitoes were treated with double-stranded RNA (dsRNA) to induce gene silencing. Following dsRNA synthesis targeting the nine *TEP*s, individual microinjections were administered to *Culex quinquefasciatus*. Seventy-two hours after this treatment, the mosquitoes were challenged with JEV, and the viral load was evaluated 3 days post-infection (dpi; Figure 1B). Unexpectedly, the knockdown of 3 *TEP* genes led to a notable decrease in JEV load in mosquitoes compared to the *GFP* dsRNA control. We also assessed the expression levels of *TEP*s after gene knockdown, observing a 3 to 5-fold reduction in the expression of the three genes (Appendix A). This reduction suggests a correlation between *TEP* knockdown and the impairment of JEV infection.

### 2.2. CqTEP27 Facilitates JEV Infection of C. quinquefasciatus

During our functional screen in vivo, silencing *CqTEP27* (NCBI: EDS28245.1) resulted in the most significant reduction in JEV load and demonstrated high specificity, prompting its selection for further investigation. The expression of *CqTEP27* decreased approximately 3-fold from 3 to 6 days post dsRNA microinjection compared to the control group (Appendix A). After treating with dsRNA for three days, JEV was inoculated into *C. quinquefasciatus,* and the viral burden was quantified using RT-qPCR at 3 and 6 dpi. The silencing of *CqTEP27* resulted in a significant reduction in JEV burdens (Figure 2A,B). Additional experiments were conducted to validate the functional role of *CqTEP27.* As *Cq*TEP27 is a secreted protein, we generated polyclonal antibodies targeting the C-terminal domain in mice using recombinant antigens expressed in *Escherichia coli* and subsequently validated through immunoblotting (Appendix A). To evaluate the effect of *Cq*TEP27 antisera on JEV infection, we injected serial dilutions of *Cq*TEP27 antisera along with JEV into mosquitoes. When compared to the mosquitoes treated with pre-immune sera, those given *Cq*TEP27 antisera showed a two-fold decrease in JEV levels at both 3 and 6 dpi, specifically in the 1:10 and 1:100 dilution groups (Figure 2C,D). These findings demonstrate that blocking *Cq*TEP27 effectively suppresses JEV infection in *C. quinquefasciatus*.

### 2.3. Interrupt CqTEP27 Impairs JEV Midgut Infection in C. quinquefasciatus

Arboviruses are maintained in a transmission cycle between mosquito vectors and mammalian hosts, with the virus infecting the mosquito midgut following a blood feeding from a viremic mammal. Overcoming midgut barriers is essential for viral acquisition and subsequent transmission [33]. Interrupting this acquisition process may decrease the infection of mosquitoes and aid in JEV prevention. The data presented herein indicate that *Cq*TEP27 acts as a susceptibility factor for JEV infection. Tissue distribution analysis of TEPs in female *Culex* mosquitoes revealed that TEP27 expression was highest in the salivary glands, with substantial levels also maintained in the midgut (Appendix A). This expression pattern suggests a potential role for *Cq*TEP27 in blood-feeding processes and midgut barrier function. Consequently, we hypothesize that targeting *Cq*TEP27 with antibodies may disrupt JEV infection during a blood meal. To test this hypothesis, we performed in vitro membrane feeding assays using a mixture of JEV-infected C6/36 cell supernatant, fresh mouse blood (50% *v*/*v*), and serially diluted *Cq*TEP27 antisera (Figure 3A). Pre-immune sera was serially diluted and served as the control, and JEV infectivity in mosquitoes was quantified via RT-qPCR at 8 dpi. Strikingly, while pre-immune sera yielded infection rates of 68–80%, *Cq*TEP27 antisera reduced infectivity to ~20% (Figure 3B,C). Similarly, following effective *CqTEP27* knockdown via dsRNA, blood feeding experiments demonstrated a substantial decrease in infection prevalence (93% compared to 37%) (Figure 3D,E and Appendix A). Plaque assays confirmed that pre-incubation with *Cq*TEP27 antisera at the concentration used in functional assays had no effect on JEV titers in vitro, which provides compelling evidence that the observed antiviral effect is specifically mediated by antibodies targeting *Cq*TEP27, rather than by non-specific factors (Appendix A). Together, these findings identify *Cq*TEP27 as a critical regulator of JEV vector competence, with targeted disruption leading to a 2–4-fold reduction in midgut infection rates, offering a potential avenue for blocking JEV spread at the vector level.

### 2.4. Immune-Blockade Against CqTEP27 in A129 Mice Prevents JEV Acquisition by C. quinquefasciatus

To investigate the role of *Cq*TEP27 antisera during JEV acquisition in vivo, we employed Type I interferon receptor-deficient (A129) mice, a well-established model that develops sustained viremia upon JEV infection and is widely used in arboviral transmission studies [34]. Mice were primarily immunized intraperitoneally (i.p.) with anti-*Cq*TEP27 or pre-immune sera, followed by JEV challenge at 1 h post-immunization. A booster dose was administered at 24 h post-primary immunization. Notably, passive immunization with anti-*Cq*TEP27 sera did not significantly affect JEV viremia levels in A129 mice, with peak viral loads observed at 4 dpi, as measured by qPCR (Figure 4A) and plaque assays (Figure 4B). To assess transmission efficiency, infected mice, with antiserum titer of 1:625,000, were exposed to mosquitoes for blood feeding at peak viremia (4 dpi, Appendix A). Following an additional 8-day incubation period under standard rearing conditions, mosquitoes receiving blood meals from antisera-treated mice exhibited a remarkable approximately 50% reduction in infection prevalence (40% vs. 75% in controls; Figure 4C,D). These findings demonstrate that while *Cq*TEP27 neutralization does not affect systemic JEV replication in murine hosts, it significantly impairs viral acquisition during mosquito blood feeding. This mosquito-specific effect positions *Cq*TEP27 as a promising target for transmission-blocking interventions against JEV.

### 2.5. CqTEP27 Suppresses AMP Expression to Facilitate JEV Infection

To clarify the role of *Cq*TEP27 in promoting JEV replication, we performed RNA-seq analysis on whole mosquitoes microinjected with JEV mixed with either anti-*Cq*TEP27 sera or pre-immune sera at 3 and 6 dpi. Volcano plot analysis indicated an upregulation of *DEFA*, *DEFC*, and *CECA* expression at both time points, while *CECA2* and *CECB* exhibited time-specific induction, with *CECA2* upregulated at 3 dpi and *CECB* at 6 dpi (Figure 5A,B). Quantification of JEV RNA in the RNA-seq samples revealed that blocking *Cq*TEP27 significantly reduced viral load (Appendix A). This result allows us to interpret the AMP upregulation. The reduced viral load in the experimental group makes it less likely that the AMP induction is merely a secondary consequence of a higher viral burden. Instead, it suggests a more complex and integrated mechanism: *Cq*TEP27 blockade may directly or indirectly prime the mosquito’s immune system, boosting AMP expression that subsequently reduces the viral replication.

These patterns of differential AMP expressions were further validated using RT-qPCR. Notably, treatment with anti-*Cq*TEP27 sera significantly increased the mRNA levels of *DEFA* and three *CECs* at both time points. While these findings generally corroborated the RNA-Seq data indicating AMP induction, some temporal discrepancies were observed: unlike the transcriptomic results, which showed time-specific induction of *CECA2* and *CECB*, qPCR detected their sustained upregulation across both days (Figure 5C,D). To investigate whether this induction is integral to the antiviral response, we co-silenced *CqTEP27* together with one of the most strongly induced *AMPs, CECA* (or *DEFA*), and then measured JEV viral loads. While silencing *CqTEP27* alone potently inhibited JEV infection, the concurrent knockdown of either *CECA* or *DEFA* partially but significantly restored JEV infection levels. These results demonstrate that the induction of *CECA* and *DEFA* is a necessary component of the antiviral pathway triggered by *CqTEP27* blockade (Figure 5E and Appendix A). Additionally, the expression of various other mosquito immune genes was also altered, suggesting the potential involvement of additional immune pathways. In summary, these results demonstrate that *Cq*TEP27 may modulate multiple AMPs to facilitate JEV replication, although further investigation required to reconcile the exact temporal patterns observed in qPCR and transcriptomic data.

### 2.6. The Vaccine-Induced Immunity of CqTEP27 Disrupts the Transmission Cycle of JEV in C. quinquefasciatus

Our findings indicate that modulating *Cq*TEP27 function effectively reduces *C. quinquefasciatus* infection rates, suggesting a potential strategy to curb Japanese encephalitis transmission in endemic regions. Consequently, vaccination with *Cq*TEP27 could prevent *C. quinquefasciatus* from acquiring JEV from infected pigs, which would lower the number of infected mosquitoes and curb further dissemination of JEV. To evaluate whether immunization against *Cq*TEP27 could block mosquito acquisition of JEV from viremic hosts, we immunized A129 mice with recombinant *Cq*TEP27, using BSA-immunized mice as controls (Figure 6A). While robust anti-*Cq*TEP27 antibody production was confirmed in immunized mice (Appendix A), viremia levels showed no significant differences across the groups following JEV challenge (5 × 10^3^ pfu, i.p.; Figure 6B,C). However, when mosquitoes fed on these infected mice at peak viremia (4 dpi), a time when the mice still exhibited high antiserum levels (titer of 1:3,125,000, Appendix A), we observed a striking approximately three-fold reduction in mosquito infection rates in the *Cq*TEP27-immunized group compared to controls after an additional 8-day incubation period (Figure 6D,E). These results clearly demonstrate that while *Cq*TEP27 immunization does not reduce disease severity in JEV-infected mice, it significantly reduces the vector infection rate after blood feeding on a host with JEV viremia, positioning *Cq*TEP27 as a promising antigen for future vaccine development.

## 3. Discussion

TEPs are essential to vertebrate innate immunity, serving as complement factors (C3/C4/C5) for pathogen clearance and as protease inhibitors (A2M) that neutralize virulence factors [21]. Beyond their immune functions, these proteins are critical for regulating immune response and maintaining host serum protease homeostasis [35]. Although insects lack immunoglobulin-based adaptive immunity, their complement-like systems exhibit remarkable functional sophistication. Current research has established critical immune roles for TEPs across major dipteran vectors: In *Anopheles* mosquitoes, *Ag*TEP1 opsonizes both Gram-negative and Gram-positive bacteria and binds to plasmodium surfaces to trigger melanization, thereby reducing oocyst numbers [21,27,28]. The *Drosophila* TEP family (*Dm*TEP2/3/6) similarly facilitates pathogen recognition and phagocytosis of both bacterial classes and fungi [16]. Particularly relevant to arboviral transmission, in *Ae. Aegypti*, *Aa*MCR specifically binds to DENV through interaction with the scavenger receptor *Aa*SR-C, subsequently activating antimicrobial peptide production to restrict viral replication [22]. However, the immune roles of TEPs in *Culex* mosquitoes, key vectors of encephalitic arboviruses, are still poorly understood.

Through functional screening, we have identified *Cq*TEP27 in *C. quinquefasciatus* as a key proviral factor that enhances JEV infection. Mechanistically, we demonstrated that *Cq*TEP27 promotes viral replication by suppressing multiple AMPs. Although we could not confirm *Cq*TEP27 knockdown at the protein level, which may due to the extensive proteolytic processing and conformational changes in functional TEPs that mask epitopes [23,36,37], this does not undermine the robustness of our functional conclusions. The consistent reduction in JEV susceptibility using two independent methods (transcriptional knockdown and antibody blockade) strongly points to a successful disruption of *Cq*TEP27 function. Thus, our phenotypic data provide compelling evidence that *Cq*TEP27 facilitates JEV infection, even as future work aims to directly quantify the protein with conformation-specific antibodies. This finding suggests that, as the principal JEV vector, *Culex* mosquitoes may have evolved specific TEP-mediated mechanisms to support viral persistence. Our data position *Cq*TEP27, and potentially other *Cq*TEP family members, as critical host factors enabling arboviral infection in this medically important vector species. Further research will investigate whether this proviral function extends to other clinically significant arboviruses transmitted by *Culex* mosquitoes, which impose significant disease burdens on humans, including Getah virus (GETV), Tembusu virus (TMUV), and related emerging pathogens.

The TEP family is characterized by its conserved GCGEQ signature sequences, which form an intrachain cysteinyl-glutamyl thioester bond between the second cysteine residue and the fifth glutamine residues [14]. This structural feature enables conformation-sensitive binding interactions that are vital for their functional attributes [19]. Mutations in the thioester bond motif disrupt covalent attachment to microbial surfaces, thereby eliminating microbial binding ability [14]. In key components of the complement system, the GCGEQ motif in human C3/C4 proteins serves as the structural basis for C3b/C4b deposition and complement cascade amplification, thereby initiating the complement effector pathway [17,38,39]. In invertebrates, the GCGEQ motif in *Anopheles* TEP1 is required for mediating anti-malarial responses, and the covalent binding event can further trigger downstream melanization or phagocytic pathways [22,23,40]. Our phylogenetic analysis revealed that among 35 thioester-containing proteins from three mosquito species and *D. melanogaster*, only 17 retain classical thioester motifs (Figure 1A; dotted gene names mark classical thioester motifs). In *C. quinquefasciatus*, we identified nine TEP homologs, six of which contain conserved thioester motifs, including *Cq*TEP3, the ortholog of *Aa*TEP3, which is known as an effector in Toll and md pathways [41]. Notably, *Cq*TEP23B contained the conserved thioester motif, while *Cq*TEP27 harbored a GCC(G)EQ mutation. We speculate that the G-to-C mutation in *Cq*TEP27 disrupts the canonical thioester bond formation, thereby enabling an alternative functional mechanism. This model is supported by other TEPs, such as *Ag*TEP3, *Dm*MCR, and *Aa*MCR, which perform immune functions despite lacking an intact thioester domain [25,42,43,44]. Transcriptomic analysis revealed that antibody-mediated inhibition of *Cq*TEP27 not only significantly regulated several *AMP* genes but also modulated multiple additional genes. Our analyses reveal that the majority of *TEP* transcripts in female mosquitoes are most abundant in the salivary glands and hemolymph, suggesting that the family likely exerts immune regulation during blood feeding or viral infection (Appendix A). This distribution pattern is similar, though not identical, to that of *Anopheles gambiae Ag*TEP1, which is predominantly restricted to the hemolymph. Given that the activation of TEP family proteins entails conformational changes that expose thioester bonds for substrate binding [21,45,46], future research will concentrate on purifying *Cq*TEP27 to elucidate its activation dynamics and to characterize the temporal regulation of the identified gene networks across arboviruses. While our functional data are currently specific to *Culex quinquefasciatus*, the known conservation of thioester-containing protein function across *Anopheles gambiae*, *Anopheles stephensi*, and *Anopheles quadriannulatus* suggests that the role of TEP27 may extend to other key JEV vectors [23,47,48]. This hypothesis of conserved function, particularly in species like *Culex tritaeniorhynchus*, represents a critical direction for our future validation.

Control strategies for vector-borne diseases are increasingly focusing on disrupting pathogen life cycles to block environmental transmission [49]. Building on the success of vaccines targeting the malaria Pfs25/Pvs25 gametocyte proteins of malaria, which reduce parasite survival in mosquitoes, researchers have identified several vector-derived factors as promising targets [50,51]. Notably, antibodies targeting the *Anopheles* midgut protein aminopeptidase (*Ag*APN1), which is a conserved ligand for *Plasmodium ookinetes*, have proven to be highly effective in preventing plasmodium development in the mosquito midgut [52]. Similarly, immunization against the tick gut receptor TROSPA has been found to effectively limit *Borrelia burgdorferi* colonization [53], while active immunity against mosGCTL-1 significantly reduces WNV acquisition in vector after a blood meal [54]. Notably, the Bm86-based vaccine represents the only licensed veterinary product for tick control, acting through immunization of cattle to disrupt *Boophilus microplus* tick midgut function and prevent pathogen transmission [55,56,57,58,59,60]. These advancements collectively demonstrate the potential of vector-focused interventions. Although licensed vaccines for JEV are currently available, ongoing assessment of their protective capacity against circulating genotype shifts is warranted [61,62]. Furthermore, the lack of JE-specific antivirals represents a critical therapeutic gap [63]. Considering the endemic spread of JEV, supplementary measures are essential to control the dissemination of the virus. The primary transmission cycle of JEV occurs between pigs and *Culex* mosquitoes, making this interaction particularly amenable to targeted intervention strategies [64,65]. Immunizing pigs can effectively lower the count of virus-carrying vectors. In this study, we discovered that *Cq*TEP27 may enhance JEV replication by suppressing the expression of AMPs. Notably, compared to feeding mosquitoes with pre-immune sera or *Cq*TEP27 antisera, treatment with anti-*Cq*TEP27 antisera led to a significant reduction in JEV infection in *C. quinquefasciatus*. The transmission-blocking effect was further validated in active vaccination trials using a murine model, where immunized A129 mice transmitted significantly fewer viruses to feeding mosquitoes (from 75–77% to 28–40%), although *Cq*TEP27 immunization does not affect the disease progression in mice. Considering the established role of swine as amplification hosts for JEV, these findings suggest that *Cq*TEP27-based immunization could disrupt natural transmission cycles at a key epidemiological juncture. Although initial proof-of-concept studies necessarily employed murine models due to the technical complexities associated with porcine systems, the significant reduction in vector infection rates strongly supports further development of this approach. Future research should prioritize porcine immunization trials to evaluate sustained antibody responses and cross-genotype efficacy, potentially establishing a new paradigm for breaking the enzootic transmission of JEV.

In summary, this study identified several *Cq*TEPs that facilitate JEV infection. Of particular interest, *Cq*TEP27 enhances JEV replication in mosquito vectors, potentially through downregulation of multiple AMPs, though contributions from other immune pathways cannot be excluded. The administration of *Cq*TEP27 antisera efficiently diminished JEV infection after the blood meal, suggesting its potential as a candidate for a transmission-blocking vaccine aimed at reducing JEV prevalence in mosquitoes. Future investigations will concentrate on: (1) uncovering the molecular mechanisms underlying *Cq*TEP27-mediated AMP regulation, and (2) optimizing *Cq*TEP27-targeted immunization approaches for field-applicable JEV control. This study significantly enhances our comprehension of JEV replication in *Culex* mosquitoes and provides a new way for developing strategies to prevent the dissemination of mosquito-borne viral diseases.

## 4. Materials and Methods

### 4.1. Animals, Cells, and Virus Cultures

A laboratory colony of *C. quinquefasciatus* (the Chengdu strain) was maintained and reared according to standard rearing procedures [66,67]. All experiments used 3–5 day-old female mosquitoes (JEV-negative by RT-qPCR). Group assignment was done arbitrarily during the transfer of mosquitoes from the rearing cage to the experimental cups. Female Balb/c mice (age 8 weeks) were obtained from Charles River Company (Beijing, China), while A129 mice (age 3 or 6 weeks) were gifted from the Department of Basic Medical Sciences, School of Medicine, Tsinghua University. All mice were maintained in a specific pathogen-free (SPF) environment, as previously described [68,69]. Mice were anesthetized by intraperitoneal injection of sodium pentobarbital (0.5 mg/10 g body weight) and euthanized via cervical dislocation if post-challenge weight loss exceeded 20% of baseline. BHK-21 and C6/36 cells (ATCC, Manassas, VA, USA) were cultured in Dulbecco’s Modified Eagle Medium (DMEM, Gibco, Waltham, MA, USA) containing 10% fetal bovine serum (Excell Bio, Cat. No#FND500, Shanghai, China) and 1% antibiotic-antimycotic (Gibco, Waltham, MA, USA). The genotype III JEV strain SA14 and 1# (GenBank U14163.1, PX136072.1, Bethesda, MD, USA) was amplified in C6/36 cells, and the titer was determined by plaque assay on BHK-21 cells (stock titer = 5.0 × 10^7^ PFU/mL). All infection work was performed in a BSL-2 facility. The sample size for this study was determined based on the effect size of this outcome measure observed in a pilot experiment.

### 4.2. Phylogenetic Tree Analysis of TEP

A homolog of *Aedes aegypti* thioester-containing protein (AaMCR, AAEL012267) was identified in the *Culex quinquefasciatus* genome (VectorBase) through protein sequence alignment, with cutoff thresholds set at an E-value < 1.0 × 10^−5^ and an S-value > 100. We confirmed its identity as a thioester-containing protein through domain architecture analysis (SMART) and phylogenetic analysis. The protein sequences used in this investigation were aligned with ClustalW as it was implemented in MEGA11 and were downoad from VectorBase (https://vectorbase.org/vectorbase/app, accessed on 22 April 2025). The JTT substitution model was used to construct a Maximum Likelihood phylogeny, and 1000 bootstrap repetitions were used to assess branch support. The sequences utilized for phylogenetic reconstruction are listed in Appendix A and have been submitted to ScienceDB.

### 4.3. Gene Silencing and Viral Challenge in Mosquitoes

The dsRNA was synthesized using the MegaScript™ T7 Transcription Kit (Invitrogen, Waltham, MA, USA) and purified using the MEGAClear™ Kit (Invitrogen, Cat. No#AM1908, USA). Cold-anaesthetised female mosquitoes (3–5 days post-eclosion) received dsRNA (300 nL) intra-thoracically using a Nanoject II injector (Drummond, Broomall, PA, USA). After 3 days of recovery, mosquitoes were challenged with 10 MID_50_ of JEV (300 nL). Viral load and knockdown efficiency were quantified via RT-qPCR at 3 and 6 dpi. The primers for dsRNA synthesis are listed in Appendix A and have been submitted to ScienceDB.

### 4.4. Membrane Blood-Feeding Assay

Mouse blood was collected, and heat-inactivated plasma (55 °C, 60 min) was recombined with washed erythrocytes (washed three times with PBS). JEV-infected C6/36 supernatant was mixed with either anti-*Cq*TEP27 sera or pre-immune sera and brought to 700 µL with DMEM. Mosquitoes were offered the mixture via membrane feeders (Hemotek, Accrington, Lancashire, UK) for 30 min. Fully engorged females were maintained for 8 days under standard conditions before sacrifice.

### 4.5. Prokaryotic Expression and Antisera Production

The C-terminal region of *Cq*TEP27 (403-971aa) was inserted into pET-28a(+) and expressed in *E. coli* BL21 (DE3) (Transgen, Beijing, China) as inclusion bodies. The inclusion bodies were solubilized in 8 M urea and purified by cobalt-ion affinity chromatography (Clontech, Mountain View, CA, USA). After loading the *E. coli* lysate onto the matrix, the resin was washed with 10 column volumes of 1% (*v*/*v*) Triton X-114 in 8 M urea to remove bound lipopolysaccharides (LPS), followed by 5 column volumes of urea buffer without Triton X-114 to eliminate detergent residues [70]. The target protein was eluted with 8 M urea buffer containing 200 mM NaCl, then concentrated and buffer-exchanged using ultrafiltration. The purified protein was quantified by BCA assay (Thermo Fisher, Waltham, MA, USA), and its purity was assessed by SDS-PAGE with Coomassie blue staining and Western blot using an anti-His-HRP antibody (Proteintech, Wuhan, China). Furthermore, the endotoxin level was determined by the Limulus Amebocyte Lysate (LAL) assay and confirmed to be below 0.1 EU/mg. The molecular weight of the protein was assessed using PageRuler Prestained Protein Ladder as a marker (Thermo Fisher, Waltham, MA, USA). Female BALB/c mice at eight weeks of age *(n* = 6) were immunized (i.p.) on days 0, 14, and 28 with 80 µg purified protein emulsified in Freund’s complete (prime) or incomplete (boosts) adjuvant. Sera were collected 14 days after the final boost, pooled, aliquoted, and stored at −80 °C.

### 4.6. Plaque Assay

BHK-21 was cultured in 6-well or 12-well plates. Diluted mouse plasma was adsorbed onto confluent BHK-21 monolayers (37 °C, 2 h). After removal of inoculum and PBS wash, cells were overlaid with 2% FBS-DMEM containing 1% agarose. After 4–5 days of incubation, the plaques were subjected to fixation with 10% formalin, staining with 0.1% crystal violet, and final quantification. The titers are expressed as mean PFU/mL ± SEM (*n* = 3).

### 4.7. RNA Extraction and RT-qPCR

Only healthy mosquitoes with intact bodies were processed. Total RNA was isolated from the entire mosquitoes and the murine blood sample using the Total RNA Kit (Axygen, Union City, CA, USA) and reverse-transcribed with the cDNA synthesis kit (TransGen, Beijing, China). RT-qPCR was performed using CFX Connect Real-Time PCR Detection System (Bio-Rad, Hercules, CA, USA). The primers and probes in this study are detailed in Appendix A, which has been deposited in the ScienceDB database. Expression levels of target genes were normalised to mosquito actin. A JEV E/actin ratio threshold of <0.001 was used to define uninfected mosquitoes, as this cutoff has been extensively validated in previous studies for reliably distinguishing infected from uninfected individuals [71,72,73,74].

### 4.8. Enzyme-Linked Immunosorbent Assay

The 96-well microplates were first coated with 1 µg *Cq*TEP27-C protein per well by overnight incubation at 4 °C. A 1 h blocking step was then performed with 5% BSA at room temperature. After that, the plates were incubated for 2 h at room temperature with serially diluted mouse sera. After PBS-T washes, HRP-conjugated anti-His antibody was added (1 h, RT). Color was developed with TMB (SeraCare, Milford, MA, USA) substrate and absorbance read at 450 nm on a microplate reader.

### 4.9. RNA-Seq Analysis of Mosquitoes

Mosquitoes were microinjected with JEV premixed with anti-*Cq*TEP27 sera or pre-immune sera (both at a 1:10 dilution, the latter serving as control). Whole bodies from 8 mosquitoes per replicate, which were independently reared and infected, were collected at 3 and 6 dpi for RNA extraction with three biological replicates per group. Total RNA was extracted using TRIzol reagent (Thermo Fisher, Waltham, MA, USA), and RNA integrity was assessed with an Agilent Bioanalyzer 2100 using the RNA 6000 Nano LabChip Kit (Agilent, Santa Clara, CA, USA). Only samples with an RNA Integrity Number (RIN) greater than 7.0 were used for library construction. cDNA was synthesized using SuperScript™ II Reverse Transcriptase (Invitrogen, Cat. No#1896649, USA), and sequencing was conducted on an Illumina NovaSeq™ 6000 platformby LC Bio Technology Co., Ltd. (Hangzhou, China), which also provided commercial RNA-seq and data analysis services in this study. Following sequencing, standard quality control metrics were assessed, including a Q30 score of over 95% for all samples, confirming the high quality of the sequencing data used for downstream analysis. Differential expression analysis was performed using the DESeq2 package, with the thresholds set at |fold change| ≥ 2 and *p*-value < 0.05. Further bioinformatic analysis was conducted using the OmicStudio tools available at https://www.omicstudio.cn/tool (accessed on 30 May 2025). The raw sequencing data generated in this study have been deposited in the National Center for Biotechnology Information (NCBI) Short Read Archive database (accession number: PRJNA1304234.)

### 4.10. Passive Immunization of A129 Mice with Anti-CqTEP27 Antisera for Blocking JEV Transmission

Eight-week-old A129 mice were primarily immunized (i.p.) with anti-CqTEP27 or pre-immune sera, followed by JEV challenge at 1 h post-immunization. Each experimental group consisted of five mice, with each mouse being an experimental unit. A booster dose was administered at 24 h post-primary immunization. Blood or plasma was collected every 24 h post-infection by tail-nick bleeding for subsequent RNA extraction or plaque assays. Beginning 24 h post-infection, each mouse was exposed daily to mosquitoes that had been starved for 24 h. Collection of fully engorged mosquitoes was performed after an 8-day incubation under standard conditions, and infectivity was evaluated via RT-qPCR.

### 4.11. CqTEP27-Based Active Immunization of A129 for Blocking JEV Transmission

Three-week-old female A129 mice were actively immunized with either purified *Cq*TEP27 or BSA according to the protocol described in “Prokaryotic expression and antisera production.” Each experimental group consisted of six mice, with each mouse being an experimental unit. A booster dose was given two weeks later, and 2 weeks after the twice booster, the mice were challenged (i.p., 5000 PFU per mouse). Beginning 24 h post-infection, each mouse was exposed daily to mosquitoes that had been starved for 24 h. The method used to detect the mosquito infection status was the same as that used in the passive immunity study.

### 4.12. Statistical Analysis

Animals were randomly assigned; mosquitoes dying before collection were excluded, and researchers were blinded to group allocation. All statistical analyses were performed using GraphPad Prism (version 9.0). Specific tests and justifications are detailed below:

Test Selection: The Shapiro–Wilk test indicated non-normality (*p* < 0.05) for most quantitative datasets (including viral loads), leading to the use of non-parametric tests. Fisher’s exact test was used for infection rates due to small sample sizes or expected frequencies below 5.

Data Presentation: Results from the Mann–Whitney test are presented as mean ± SEM, while data analyzed by Fisher’s exact test (e.g., infection rates) are presented as mean values. The Cliff δ values and 95% confidence interval (CI) for the main figure are provided in Appendix A.

Multiple Comparisons Correction: For multiple comparisons, the FDR was controlled using the BH method, and the results of this correction are indicated in the figure legends.

Definition of Replicates: Each data point in the figure represents an independent biological replicate, defined as an individual mosquito from a distinct egg batch or a mouse from a separate litter. Technical replicates were averaged into a single value per biological replicate and excluded from statistical tests. Experiments were repeated at least twice, with sample sizes for key studies justified by effect sizes from pilot experiments.

Endpoints: The primary endpoints for the study were defined as viral load (quantified by RT-qPCR) and infection rate (percentage of infected individuals per group) at specified dpi.

## Figures and Tables

**Figure 1 ijms-26-11727-f001:**
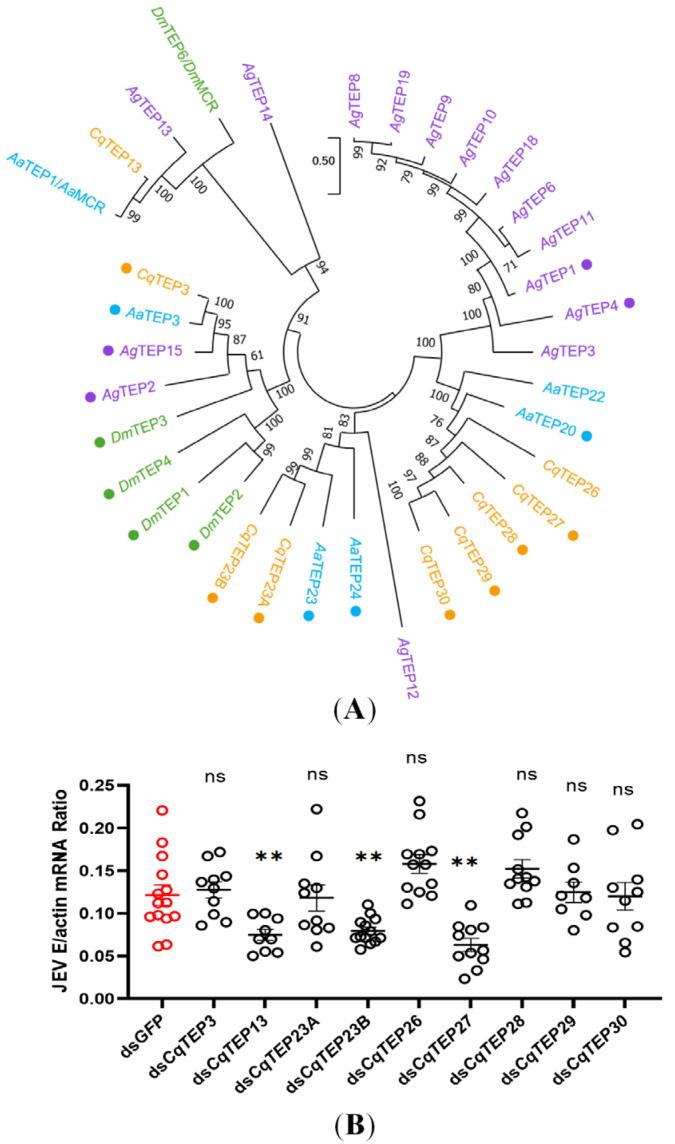
(**A**) A Maximum Likelihood (ML) phylogenetic tree was reconstructed from an alignment of 35 TEP sequences. The tree was inferred using MEGA11 with 1000 bootstrap replicates to assess node support. *An. gambiea* (*Ag*), *Ae. aegypti* (*Aa*), *D. melanogaster* (*Dm*), and *C. quinquefasciatus* (*Cq*) are indicated with purple, blue, green, and orange, respectively. The period represents the iTEPs with TE motif (GCGEQ) domain. (**B**) Effect of individual *CqTEP* silencing on whole-body JEV load. At 72 h post-injection with dsRNA targeting *CqTEP* genes (or *dsGFP* control), mosquitoes were challenged with 10 times the Median Infective Dose (MID_50_) of JEV. Whole-body viral RNA loads were quantified by real-time quantitative polymerase chain reaction (RT-qPCR) at 3 dpi. (Control, *n* = 14; Treatment, *n* = 12). Statistical significance is indicated as follows: ** *p* < 0.01; ns, not significant.

**Figure 2 ijms-26-11727-f002:**
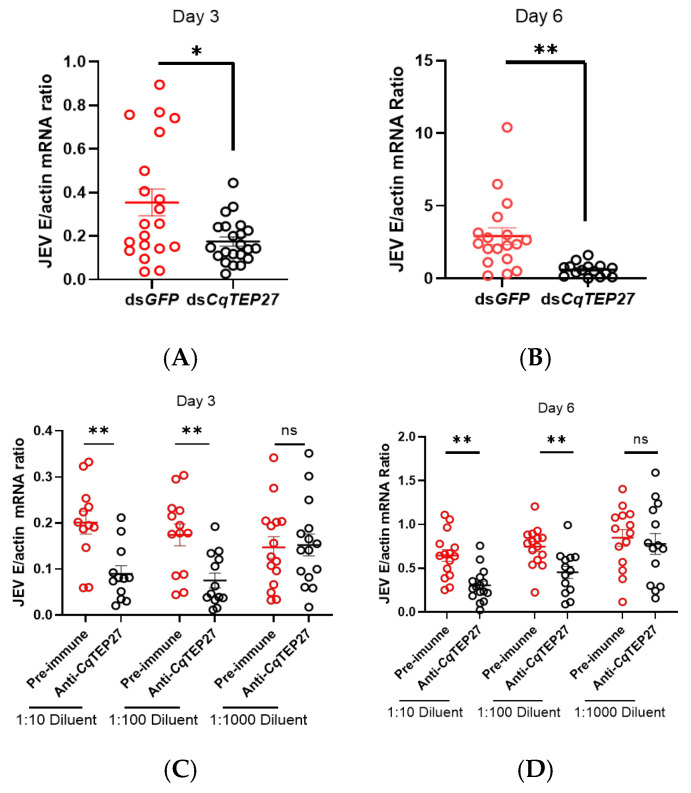
(**A**,**B**) Effect on JEV load after *CqTEP27* silencing and micro-injection challenge at 3 dpi (**A**) (*dsGFP*, *n* = 20; *dsTEP27*, *n* = 22) and 6 dpi (**B**) (*dsGFP*, *n* = 18; *dsCqTEP27*, *n* = 15). Female mosquitoes were injected with *CqTEP27*-targeting dsRNA (or *dsGFP* control). Following injection, mosquitoes were exposed to 10 MID_50_ of JEV after 72 h. JEV load was quantified via RT-qPCR. (**C**,**D**) Effect on JEV load after neutralization of *Cq*TEP27 by microinjection at 3 dpi (**C**) (1:10, *n* = 12; 1:100, *n* = 13; 1:1000, *n* = 15) and (**D**) (Pre-immune, 1:10, *n* = 15, 1:100, *n* = 16, 1:1000, *n* = 14; Anti-*Cq*TEP27, 1:10, *n* = 16; 1:100, *n* = 13; 1:1000, *n* = 14) 6 dpi. A serially diluted anti-*Cq*TEP27 polyclonal antibody was mixed with SA14-infected cell supernatant and micro-injected into mosquitoes; JEV load was quantified via RT-qPCR. All findings significant at *p* < 0.05 remained statistically significant following Benjamini–Hochberg (BH) correction for multiple comparisons (false discovery rate (FDR) < 0.05). Statistical significance is indicated as follows: ** *p* < 0.01; * *p* < 0.05; ns, not significant.

**Figure 3 ijms-26-11727-f003:**
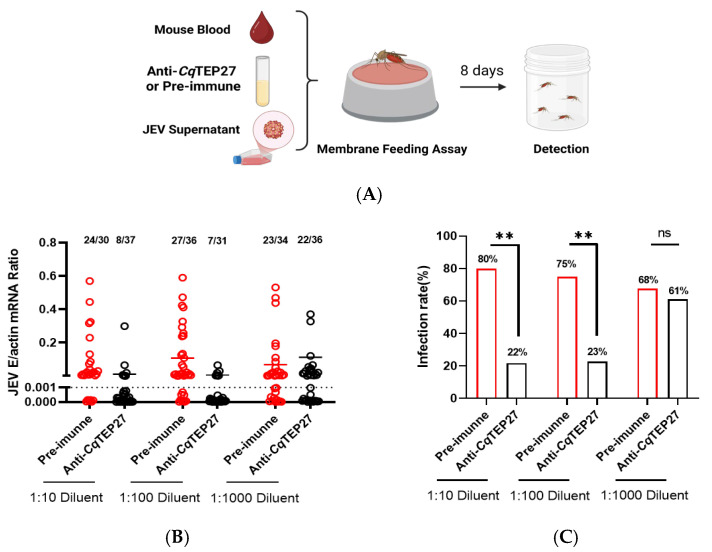
(**A**) Schematic of artificial membrane feeding assay (Created in Biorender. Huang, Y., et al. (2025) https://BioRender.com). (**B**,**C**) Effect on JEV infection load (**B**) and infection rate (**C**) following *Cq*TEP27 neutralization via membrane feeding challenge. Mosquitoes were challenged via membrane feeding with either anti-*Cq*TEP27 antibody or pre-immune sera, and the infection prevalence was determined at 8 dpi by RT-qPCR. All findings significant at *p* < 0.05 remained statistically significant following BH correction for multiple comparisons (FDR < 0.05). (**D**,**E**) Effect of *CqTEP27* silencing followed by membrane feeding challenge on JEV infection load (**D**) and infection rate (**E**). Female mosquitoes were injected with 300 nL ds*CqTEP27* (or ds*GFP* as control) and three days later, they received an infectious blood meal consisting of SA14-infected cell supernatant mixed with fresh mouse blood. Infection status was determined at 8 dpi by RT-qPCR using the same threshold. (**B**,**D**) Numbers above each bar indicate infected/total mosquitoes, and the exact *n* values are indicated above the graph bars. (**C**,**E**) The data are presented as the percentage of infected mosquitoes. Statistical significance is indicated as follows: ** *p* < 0.01; ns, not significant.

**Figure 4 ijms-26-11727-f004:**
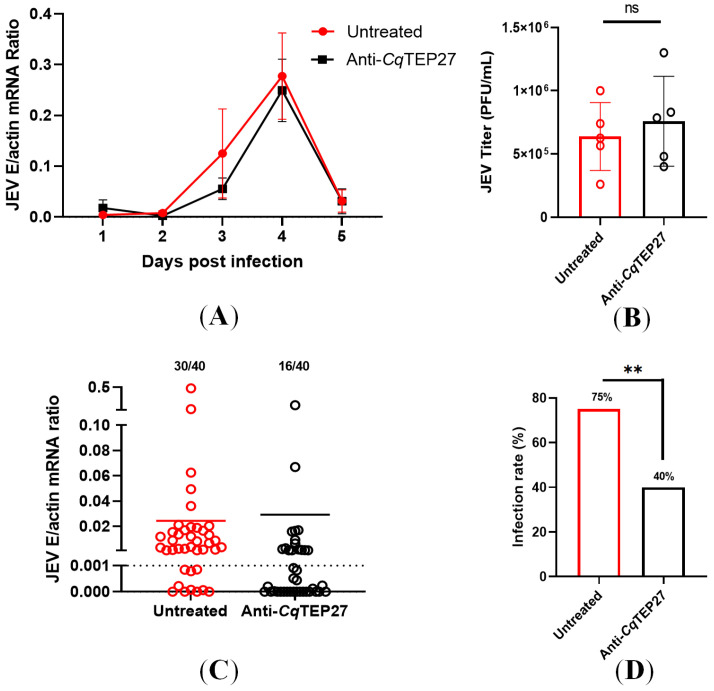
(**A**,**B**) Viremia in A129 mice following passive immunization of anti-*Cq*TEP27 or control (pre-immune) sera. (**A**) Daily JEV load in blood of A129 mice following passive immunization against *Cq*TEP27 and JEV challenge. JEV load was quantified via RT-qPCR every 24 h after viral challenge (*n* = 5). Data presented as mean ± SEM. (**B**) JEV viral titer in blood of passively immunized mice peaks at 4 dpi (*n* = 5). Infectious virus titers were determined by plaque assay during maximum viremia. Each data point represents an individual mouse. (**C**,**D**) Effect of host passive immunization against *Cq*TEP27 on mosquito JEV infection load (**C**) and infection rate (**D**) post-blood meal. Mosquitoes were allowed to feed on infected mice at 4 dpi, and whole-body JEV RNA was quantified 8 dpi. (**C**) Numbers above each bar indicate infected/total mosquitoes; the exact n values are indicated above the graph bars. (**D**) The data are presented as the percentage of infected mosquitoes. Statistical significance is indicated as follows: ** *p* < 0.01; ns, not significant.

**Figure 5 ijms-26-11727-f005:**
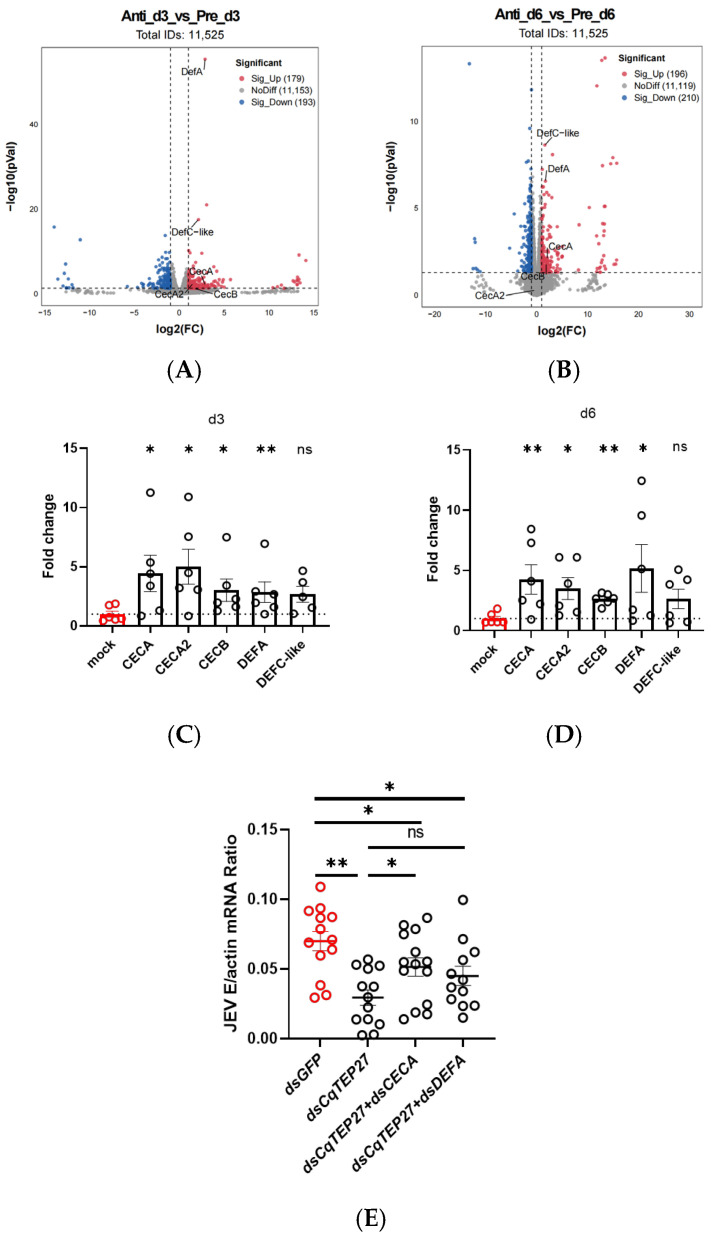
(**A**,**B**) Volcano plots of RNA-seq data from mosquitoes following challenge with SA14. Mosquitoes were microinjected with virus mixed with either anti*-Cq*TEP27 sera or pre-immune sera, and transcriptome analysis of whole mosquitoes at (**A**) 3 dpi and (**B**) 6 dpi. The dashed lines indicate the cut-off criteria for differential expression. Genes beyond the vertical lines (log_2_|FC| > 1) and above the horizontal line (*p* < 0.05) are considered significantly upregulated (right) or downregulated (left), with *AMP genes* highlighted. (**C**,**D**) Validation of *AMP gene* expression by RT-qPCR at (**C**) 3 dpi and (**D**) 6 dpi (*n* = 6). Data represent the fold-change (anti-*Cq*TEP27/pre-immune sera) in mRNA levels. After correction for multiple testing, all associations shown here have an *FDR* < 0.1, suggesting a potential biological trend. (**E**) Silencing of *CECA* or *DEFA* partially rescues the suppression of JEV replication caused by *CqTEP27* knockdown (*n* = 14). Mosquitoes were injected with gene-targeting dsRNA and then infected with JEV. Viral load was determined by RT-qPCR at 3 dpi. Statistical significance is indicated as follows: ** *p* < 0.01; * *p* < 0.05; ns, not significant.

**Figure 6 ijms-26-11727-f006:**
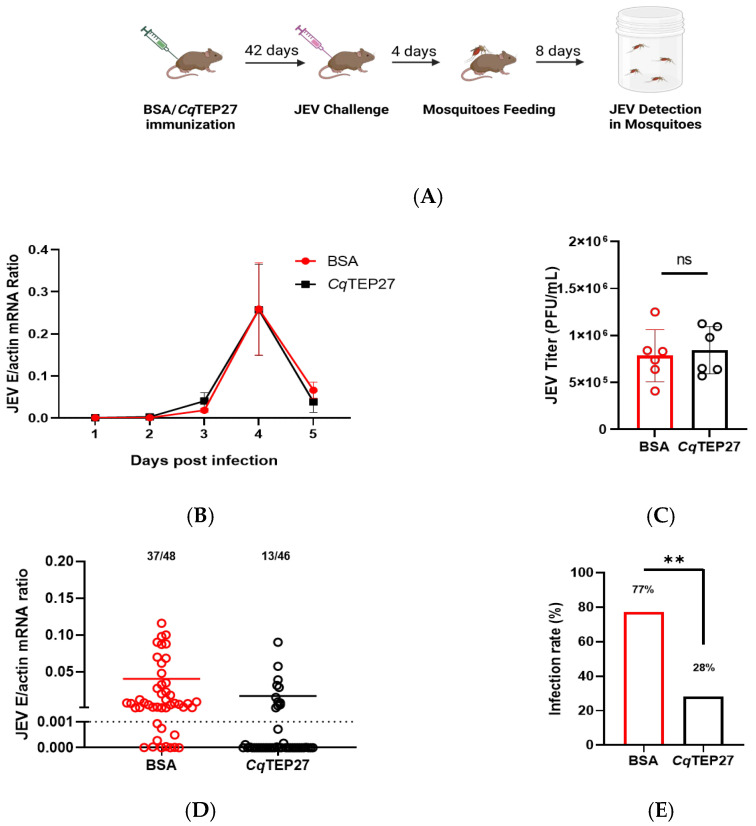
(**A**) Active immunization schedule: A129 mice received prime/boost vaccinations with *Cq*TEP27 at 2-week intervals, followed by JEV challenge at 42 days post-prime and mosquito exposure (Created in Biorender. Huang, Y., et al. (2025) https://BioRender.com). (**B**) Daily JEV load in blood of A129 mice following active immunization against *Cq*TEP27 and JEV challenge (*n* = 6). JEV load was quantified via RT-qPCR at 24 h intervals post-infection. (**C**) JEV viral titer in blood of actively immunized mice peaks at 4 dpi. Infectious virus titers were determined by plaque assay during maximum viremia (*n* = 6). Each data point represents an individual mouse. (**D**) Effect of host active immunization against *Cq*TEP27 on mosquito JEV load post-blood meal. Mosquito infection rates after following blood-feeding on immunized mice at peak viremia (4 dpi). Numbers above each bar indicate infected/total mosquitoes; each dot is one mosquito, and the exact n values are indicated above the graph bars. Data are presented as the mean. (**E**) Data are presented as the percentage of infected mosquitoes. Statistical significance is indicated as follows: ** *p* < 0.01; ns, not significant.

## Data Availability

The data that support the findings of this study are directly available in ScienceDB (https://www.scidb.cn/s/zuuQVf, accessed on 24 November 2025).

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
