# Peer review of "Thioester-Containing Protein TEP27 in Culex quinquefasciatus Promotes JEV Infection by Modulating Host Immune Function"

_ijms, 2025, doi:10.3390/ijms262311727_

Round 1

Reviewer 1 Report

Comments and Suggestions for Authors

Because JEV is a major health problem and poses a challenge to the production of food derived from pigs, the authors studied the role of the Thioester-Containing Protein TEP27 in JEV replication in the vector mosquito, Culex quinquefasciatus.

It was known that type orthologos proteins prevent viral replication in other mosquito species, such as Ae. aegipty. However, in the present study, it was widely reported that TEP27proteins with GCGEQ motifs promote JEV replication in Culex mosquitoes.

Based on this finding, the authors investigated whether silencing these genes or inactivating them with antibodies could reduce JEV replication in mosquitoes, and indeed this occurred. They demonstrated that both strategies are effective at reducing viral replication and that antibodies generated in mice can inhibit JEV replication by binding the TEP27 protein and blocking its promoter activity.

 Thus, this research paves the way for developing a new strategy to control JEV transmission by immunizing pigs, an important host for JEV. So, pigs immunized with this protein will produce antibodies that, when ingested by mosquitoes during feeding, can inhibit the JEV replication-facilitating action within mosquitoes caused by TEP27proteins with GCGEQ motifs. Reducing viral load can lower the risk of JEV transmission and infection.

I have a question regarding the JEV replication-promoting activity of this protein. Is this the same phenomenon that would be observed in other species of the Culex genus that transmit JEV, or is it only characteristic of the quinquefasciatus specie?

Reviewer 2 Report

Comments and Suggestions for Authors

See below.

Reviewer 3 Report

Comments and Suggestions for Authors

Dear authors, the report is attached. 

Comments on the Quality of English Language

The English is generally clear and understandable, but it still needs moderate professional polishing. Some sentences are long and slightly awkward, with minor grammatical inconsistencies and non-idiomatic phrasing. After light copy-editing for fluency, tense consistency, and word choice, the manuscript will fully meet publication standards.

Round 2

Reviewer 3 Report

Comments and Suggestions for Authors

Dear authors, please see attached.
